# Grain Refinement Assisted by Deformation Enhanced Precipitates through Thermomechanical Treatment of AA7055 Al Alloy

**Jinrong Zuo** [1,2,*] **, Longgang Hou** [3] **, Xuedao Shu** [1,2] **, Wenfei Peng** [1,2] **, Anmin Yin** [1,2] **and Jishan Zhang** [3]

1   College of Mechanical Engineering and Mechanics, Ningbo University, Ningbo 315211, China; shuxuedao@nbu.edu.cn (X.S.); pengwenfei@nbu.edu.cn (W.P.); yinanmin@nbu.edu.cn (A.Y.)
2   Zhejiang Provincial Key Laboratory of Part Rolling Technology, Ningbo University, Ningbo 315211, China
3   State Key Laboratory for Advanced Metals and Materials, University of Science and Technology Beijing, 30 Xueyuan Road, Haidian District, Beijing 100083, China; lghou@skl.ustb.edu.cn (L.H.); zhangjs@skl.ustb.edu.cn (J.Z.)
*   Correspondence: zuojinrong@nbu.edu.cn; Tel.: +86-0571-87600534

**Abstract:** Fine-grained sheets of AA7055 Al alloy were produced by an improved double-stage rolling thermomechanical treatment (DRTMT) assisted by deformation-enhanced precipitates (DEPs). The DRTMT composed of a low temperature pre-deformation, an intermediate annealing, and a final hot rolling exhibited significantly superior tensile ductility to the conventional rolling thermomechanical treated alloy (CRTMT). Numerous fine spherical DEPs appeared after the pre-deformation. Those DEPs could exert a strong drag force on the migration of boundaries and dislocations. Dislocation cells were formed due to the drag force and dynamic recovery. These dislocation cells become polygon sub-grains by static recovery during the subsequent intermediate annealing. After the final hot deformation, with further deformation and rising temperature, low angle grain boundaries gradually stabilized and transferred to high angle grain boundaries. Due to the transformation, fine equiaxed grains were formed after DRTMT. The DRTMT alloys display superior elongation to the CRTMT alloy while maintaining high strength for grain refinement. Thus, DRTMT would be a good alternative to manufacture different heat-treatable Al alloys with fine grains efficiently.

**Keywords:** 7xxx Al alloy; rolling; precipitate; mechanical property; microstructure

## 1. Introduction

With a high strength to weight ratio, good toughness, and high corrosion resistance, 7xxx series Al alloys have been applied in many areas of industrial manufacturing (e.g., aircraft, shipbuilding, automotive and nuclear industries) [1]. Starke and Staley [2] found that further improving their properties (safety, lightweight, and fuel economy) is still greatly urgent, which are vitally charged by the formation of dispersed fine precipitates ($MgZn_2$) during the artificial aging. Bergsma et al. [3] increased the content of key alloy elements such as Mg, Zn, Cu, and more precipitates were produced, resulting in enhanced strength. However, McQueen and Celliers found that their plasticity/corrosion resistance may be deteriorated [4].

Zuo et al. [5] designed an improved thermomechanical treatment and found that grain refinement can benefit mechanical properties. Similarly, Alhamidi and Horita [6] revealed that grain refinement can meet the performance demands in aerospace industries. Fine-grained structures can be obtained by developing severe plastic deformation (SPD), e.g., equal channel angular pressing (ECAP) [7],

cryorolling [8] and improved thermal mechanical treatment (TMT) [9]. Zheng et al. [10] investigated the evolution of microstructure and strengthening of 7050 Al alloy by ECAP and proved that ECAP has its own advantages of producing stable ultra-fine grains. Panigrahi and Jayaganthan [11] studied the development of ultrafine grained 7075 Al alloy by cryorolling and found that cryorolling can suppress the dynamic recovery and formation of GP zones, and thus create high-density dislocations (accumulating more deformation storage energy for recrystallization to refine grains). However, their applications may be limited by severe plastic strains in ECAP and cryorolling, and the difficulties in producing larger scaled alloys. The conventional TMT process (combined plastic deformation and heat treatment) can improve the comprehensive properties of alloys by affecting their microstructures. Humphreys and Hatherly [12] revealed that conventional rolling process can refine grain size to 30–250 μm (hard to obtain fine-grained structures of the high stacking fault energy materials like 7xxx series Al alloys). It can be seen from the Al-MgZn$_2$ phase diagram (Figure 1) that the solubility of MgZn$_2$ in Al matrix changes greatly with temperature, so different MgZn$_2$ precipitates can be obtained through heat treatment/deformation. Zuo et al. [13] found that some particles in Al alloys may play important roles for grain refinement during TMTs. Huo et al. [14] had designed different TMTs to refine grains of 7075 Al alloy and proved that small particles can inhibit the recrystallization while Huo et al. [15] also found large particles can stimulate nucleation for recrystallization.

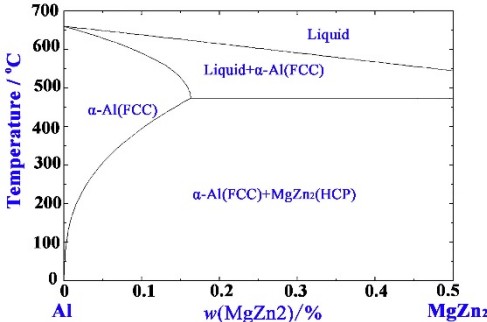

**Figure 1.** The equilibrium phase diagram of Al-MgZn$_2$ pseudo-binary system.

Waldman et al. [16] designed a multi-step TMT (Wal-TMT: homogenization, furnace cooling, warm rolling, solutionizing, and recrystallizing) to manufacture fine-grained 7075 Al alloy. The ingot was homogenized to precipitate Cr-particles at high temperature and then obtained coarse Al (Zn, Mg, Cu) particles acting as PSN (particle-stimulated nucleation) particles at a slow cooling rate. Then the ingot was deformed, recrystallized, and artificially aged, inducing equivalent strength and significantly better elongation/area reduction than conventionally processed alloy. Later, this process was partially improved by Wert et al. [17] to obtain fine-grained 7xxx Al alloy (Wer-TMT). In Wer-TMT, the step furnace cooling was replaced by 400 °C/8 h over aging. Therefore, Wer-TMT is much more time-saving and the final grain size was almost the same as Wal-TMT, resulting in higher mechanical properties and corrosion resistance.

However, it is noteworthy that the above both TMTs need a long period heat preservation treatment to acquire enough large particles for PSN effect, both energy and time consuming. Moreover, Huo [18] found it is very difficult for the thick plate to obtain enough deformation storage energy by sever rolling at low temperature (especially for 7xxx series aluminum alloys). However, PSN is not the only way to refine grains. In order to refine grains economically, a new short-cycled TMT process, DRTMT (shown in Table 1) including DEPs (deformation enhanced precipitates), was applied to manufacture 7055 Al sheets. The corresponding mechanical properties and microstructure evolution were also studied as well as the contrastive CRTMT (parameters selected according to [19]). DEPs as a substitute for the long period heat preservation treatment was used in the new DRTMT. DRTMT exhibits higher efficient, more energy-saving and time-saving. Moreover, alloys were deformed mostly at relatively high temperature during DRTMT, with lower yield strength and better formation ability.

**Table 1.** Specific parameters of TMTs.

| Process | Predeformation | IA | Final Hot Rolling | T6 Aging |
|---------|----------------|-----|-------------------|----------|
| CRTMT | - | - | 400 °C/80% 15–3 mm | |
| DRTMT$_1$ | | | 300 °C/20% 6–3 mm | |
| DRTMT$_2$ | 300 °C/60% 15–6 mm | 430 °C/5 min | 350 °C/20% 6–3 mm | 475 °C/0.5 h + 120 °C/24 h |
| DRTMT$_3$ | | | 400 °C/20% 6–3 mm | |
| DRTMT$_4$ | | | 430 °C/20% 6–3 mm | |
| DRTMT$_5$ | | | 450 °C/20% 6–3 mm | |

Note: Predeformation-1st-stage warm rolling, IA-intermediate annealing, Final hot rolling-2nd-stage hot rolling, T6 aging-solution and aging treatment.

## 2. Materials and Methods

To investigate the properties and microstructures evolution, the 15 mm thickness as hot rolled commercial AA7055 Al alloy plates 8.38Zn-2.07Mg-2.31Cu-0.092Fe-0.056Si (wt.%) were used. Which were solution treated (470 °C/16 h + 475 °C/8 h) to return to initial state (recrystallized state and most precipitates dissolved back). The solution treated plates were then rolled at 15 rpm (with two 270 mm rolling mills) by DRTMT and CRTMT (parameters in Table 1 with the details in Table 2). Both plates were heated for 10 min between passes to maintain constant rolling conditions. The deformation rate $\bar{\varepsilon}$ (s$^{-1}$) is also an important variable calculated by the Ekelund formula [20]:

$$\bar{\varepsilon} = \frac{2v\sqrt{\frac{h_0-h}{R}}}{h_0 + h_1} \tag{1}$$

where $R$ (mm) is the radius of the roll, $v$ (mm/s) is the rolling speed, and $h_0$, $h_1$ (mm) are the thickness before and after rolling.

**Table 2.** Parameters for DRTMT and CRTMT.

| Pass | Reduction (mm) | $R$ (mm) | $v$ (mm/s) | $h_0$ (mm) | $h_1$ (mm) | $\bar{\varepsilon}$ (s$^{-1}$) |
|------|----------------|----------|------------|------------|------------|--------------------------------|
| 1 | 3 | | | 15 | 12 | 2.33 |
| 2 | 3 | | | 12 | 9 | 3.01 |
| 3 | 1.5 | | | 9 | 7.5 | 2.70 |
| 4 | 1.5 | 135 | 212 | 7.5 | 6 | 3.30 |
| 5 | 1 | | | 6 | 5 | 3.32 |
| 6 | 1 | | | 5 | 4 | 4.05 |
| 7 | 1 | | | 4 | 3 | 5.21 |

Note: Pass-rolling times, Reduction-decrease of thickness of the rolled piece, $R$—radius of the roll, $v$—rolling speed, $h_0$, $h_1$—thickness before and after rolling, $\bar{\varepsilon}$—strain rate.

Specimens cut from the rolled plates in the Normal direction-Rolling direction (ND-RD) plane and then mechanically polished, and observed by Ultra 55 scanning electron microscopy (SEM, Zeiss, Jena, Germany) with an energy dispersive spectrometer (EDS) as well as optical microscopy (OM, Zeiss, MC80DX, Jena, Germany). The polished specimens were etched with Keller's reagent (1.5% HCl +1% HF +2.5% HNO$_3$ +95% distilled water (Vol.%)). The software "Image J" (version:1.52v, National Institutes of Health, MD, USA) was used to analyze sizes and area fractions of the second phase. X-ray diffraction (XRD) measurement was performed on a X-ray diffraction (D/MAX-RB, Rigaku, Tokyo, Japan) equipped with a Cu target ($\lambda$ = 0.15406 nm) operating at 40 KV for 2$\theta$ from 10° to 70° (step size 10° per min) at room temperature. Zeiss Ultra 55 SEM with electron back scattered diffraction (EBSD) and the system "Channel 5" (Oxford Instruments, Witney, Oxon, UK) were used to analyze grain sizes/shapes, grain/sub-grain orientations as well as grain distributions. In the EBSD maps, black lines represent HAGBs (high angle grain boundaries with misorientations $\theta$ > 15°) while gray lines represent the LAGBs (low angle grain boundaries with grain boundary misorientations 2°

< θ < 15°). The EBSD specimen was sampled on the ND-RD plane. After mechanically polishing, the specimen surface was electropolished with 70% methanol and 30% nitric acid solution at −30 °C and 30 V voltage to remove the deformation layer. The samples were installed on a pre-titled sample holder, with a tilt angle of 70°, an acceleration voltage of 20 kV, a step size of 1 μm, and a coverage area of about 0.12 mm². Three such random areas were examined to consider the statistics. The data was taken from the center of the plate, and the average value of three areas was taken. Subsequently, thin films (with 1 mm thick) were cut from the ND-RD section of the plate center, on which discs with diameter of 3 mm were punched, then ground to about 100 μm, and then twin-jet electropolished with 70% methanol and 30% nitric acid to prepare the thin film for transmission electron microscopy (TEM, H-800 (Hitachi, Tokyo, Japan) and Tecnai F30 (FEI, OR, USA) and temperature: −20~−30 °C, potential difference: 30 V). MTS 810 universal testing machine (MTS, MN, USA) was used to measure mechanical properties at room temperature with a nominal strain rate of $10^{-3}$ $s^{-1}$. Tensile samples were prepared according to ASTM E8-04 (gauge length 25 mm and gauge width 6 mm, dog-bone shape and axis paralleling to the rolling direction). In order to get reliable statistical results, consistent tensile/yield strength values were acquired by three parallel specimens. The fractured surfaces after tensile tests were also characterized by SEM.

## 3. Results and Discussions

### 3.1. Microstructures and Mechanical Properties of the Rolled CRTMT and DRTMT Al Alloys after T6 Aging

Figure 2 shows the microstructure of 7055sq. It can be seen that the initial grains of 7055sq are coarse (grain size in length ≥400 μm, in transverse ≥200 μm. There still exist two kinds of second phase in the alloy matrix after solution treatment: coarse hollowed fishbone like particles (Figure 2b) and coarse spherical particles (Figure 2c). EDS results in Figure 2d show that the hollowed fishbone like particles in Figure 2b are Fe-rich phase while the coarse spherical particles in Figure 2c are S ($Al_2CuMg$) phase. These particles cannot be dissolved back in the solution treatment. Dispersed $MgZn_2$ can hardly be observed in Figure 2b,c, which indicates full redissolution into the matrix during two-stage solid solution treatment.

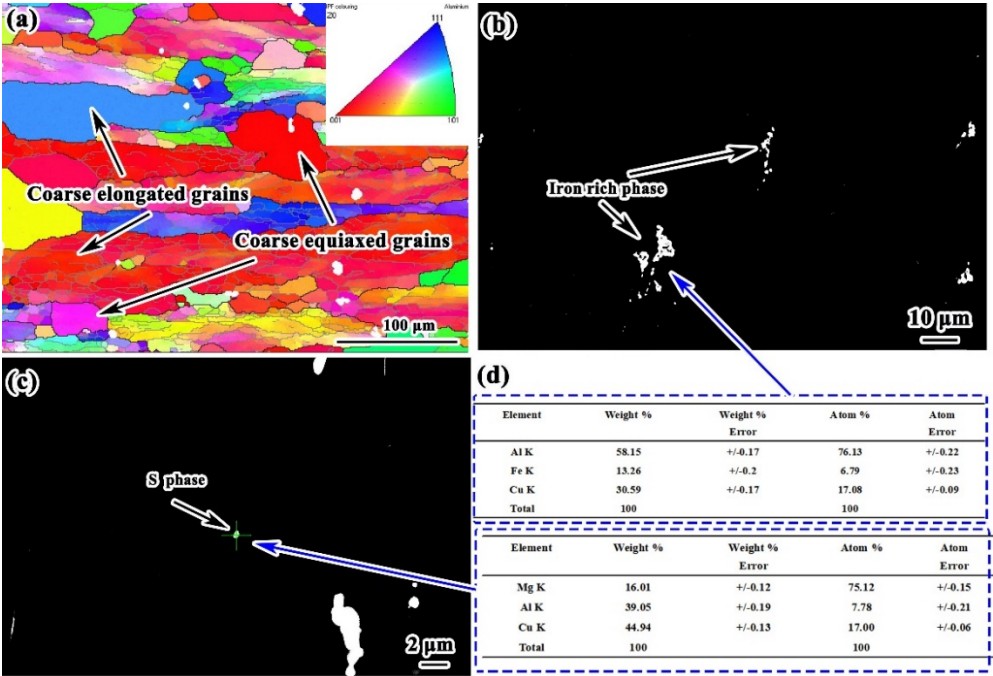

**Figure 2.** Microstructure of 7055sq: (**a**) EBSD image; (**b**) SEM for iron rich phase; (**c**) SEM for S phase; (**d**) EDS result.

Figure 3 shows recrystallized microstructures of the CRTMT and DRTMT Al alloys. CRTMT shows elongated grains of unequal width and coarse equiaxed grains (Figure 3a) while that of the latter (DRTMT$_{1\sim5}$) obtain grain refinement in various degrees. Grains in DRTMT Al alloy become wider in transverse and shorter in length compared to those in CRTMT Al alloy (>100 μm in length and 7 μm in transverse in Figure 3a), tending to be more equiaxed.

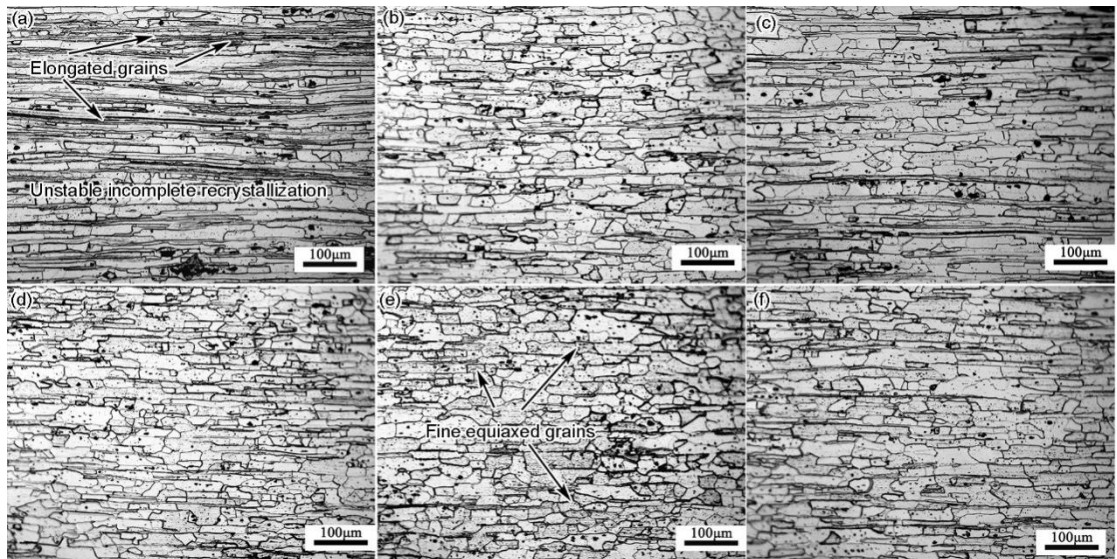

**Figure 3.** OM of the recrystallized sheets: (**a**) CRTMT; (**b**) DRTMT$_1$; (**c**) DRTMT$_2$; (**d**) DRTMT$_3$; (**e**)DRTMT$_4$; (**f**) DRTMT$_5$.

The refinement of DRTMT is better, especially DRTMT$_4$ (Figures 3d and 4a). The quantity of fine equiaxed grains of DRTMT samples is far more than CRTMT sample (Figure 3). Figures 3a–e and 4a reveal that grain sizes after recrystallization were reduced firstly and then increased with the increased second-stage deformation temperatures (DRTMT$_1$~DRTMT$_5$). The recrystallized grains are mainly affected by different TMT parameters.

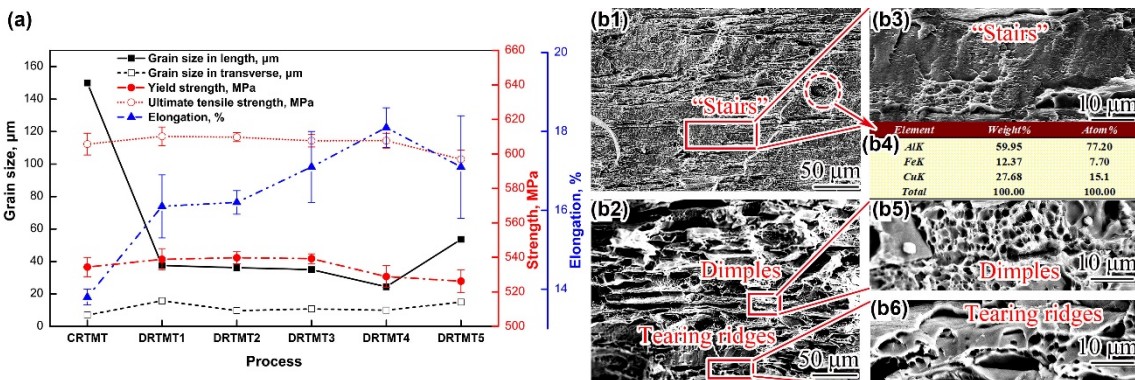

**Figure 4.** (**a**): Grain size, strength and elongation of CRTMT and DRTMT after T6 aging; Tensile fractures of CRTMT (**b1**) and DRTMT$_4$ (**b2**) and the localized morphologies ((**b3,b4**) (EDS result) for CRTMT, (**b5,b6**) for DRTMT$_4$).

After solutionizing (475 °C for 0.5 h) and T6 aging treatment (120 °C for 24 h), mechanical properties (yield/tensile strength, elongation) of all processed alloys are presented in Figure 4a. It shows that elongation of the DRTMT is higher, especially the DRTMT$_4$ sample (more than 18% in Figure 4a), which may be attributed to grain refinement (Figure 3e).

It is worth noting that the tensile/yield strengths of DRTMT alloy are similar to CRTMT alloy, but the former obtains higher elongation than the latter in T6 state (Figure 4a). Here lies the explanation for different grain sizes exhibiting similar strengths. Precipitation hardening would be the main contribution in hardening response of heat-treatable Al alloys like 7xxx series and the refined grains slightly affect the tensile (yield) strengths of 7xxx series Al alloys. Hall and Petch found that the strength σ (MPa) is closely related to the average grain diameter *d* (μm) [21,22]:

$$\sigma = \sigma_0 + kd^{-1/2} \tag{2}$$

where $\sigma_0$ (Mpa) is the lattice frictional stress. *k* (MPa·μm$^{1/2}$) is coefficient of grain boundary on strength. $\sigma_0$ and *k* are constants. Grain refinement (reduced average grain diameter *d*) should improve the yield strength σ. However, the yield strengths in Figure 4a remain nearly the same with grain size ranging. Wert [23] found that the coefficient *k* in Hall-Petch equation for T6-treated 7075 Al was just 0.12. Ma et al. [24] confirmed the high reliability of this value. But Di Russo et al. [25] designed a new thermo-mechanical procedure to improve the ductility and toughness of Al-Zn-Mg-Cu alloys and found grain refinement plays a great role in improving the ductility. The influence of grain size on dislocation pile-ups and tensile fracture was investigated by Terlinde and Luetjering [26] who found that longer total grain boundaries (fine grains) exhibit better coordination among grains during tensile tests. With fine-grained structures, the homogeneous dislocation slipping can occur and much more dislocations can be stored, making a higher strain hardening ability so as to delay fracture and contribute to the high ductility. This will increase the resistance of crack initiation and improve the ability of dislocation storage. Therefore, DRTMT treated alloys can get a significant increase of elongation under the same solution-aging treatment.

Figure 4b1–b6 shows the presence of tearing ridges, coarse intermetallic particles and dimples on the fractures of both alloys. The fracture of the CRTMT Al alloy appears to be a stair-step pattern (relatively flat in Figure 4b1,b3) with a small amount of large and shallow dimples that contain some coarse particles (e.g., iron rich phase from EDS result in Figure 4b4), which indicates that the fracture of CRTMT is brittle. However, on the fracture of DRTMT$_4$ Al alloy, the tearing ridges and large quantities of dispersed deep dimples appear throughout the fracture surface (Figure 4b2,b4,b5), indicating a relatively ductile mode which would be attributed to grain refinement.

### 3.2. Microstructure Evolution during TMTs

Tian and Wang [27] reported that plenty of second phase particles (e.g., coarse second phases (size: >0.6 μm), e.g., phases containing Fe, Si, and Cu elements), dispersed phases containing Cr/Mn/Zr (size: 0.02–0.6 μm), and fine precipitates (size: <0.5 μm) in Al alloys can play important roles during TMTs. Their interaction with the sub-structures during TMTs is one of the key issues to reveal the related grain refinement mechanism, especially the controllable precipitates. The coarse second particles and dispersed phases are hardly changed during the subsequent heat treatment and TMTs, but the precipitates can be precipitated or re-dissolved during the heat treatments/TMTs. These precipitates (e.g., MgZn$_2$ in 7055 Al alloy) were investigated with respect to their evolution and effects during the heat treatment and TMTs by Zuo [28]. Cai [29] studied the relationship between deformation and precipitation and revealed that the formation, spheroidization and refinement of precipitates can be enhanced by warm deformation which introduces defects (e.g., vacancies and dislocations) into alloys. Adequate energy/structural fluctuation and rapid diffusion paths of solute atoms can be afforded by these defects.

SEM with the embedded TEM images (the white spots are large particles that were etched away during twin-jet electro-polishing.) are shown in Figure 5. After the first stage deformation (DRTMT-60%), spherical particles (~38 nm) wrapped by dislocation cells were precipitated (Figure 5a). Dislocations were tangled together to form dislocation walls or cells of varying shape and size because of dynamic recovery. Dislocation slip was obstructed due to the drag force by precipitates.

Thus, dislocations were piled up and formed dislocation-cells rather than polygonizing to form well-ordered sub-grains.

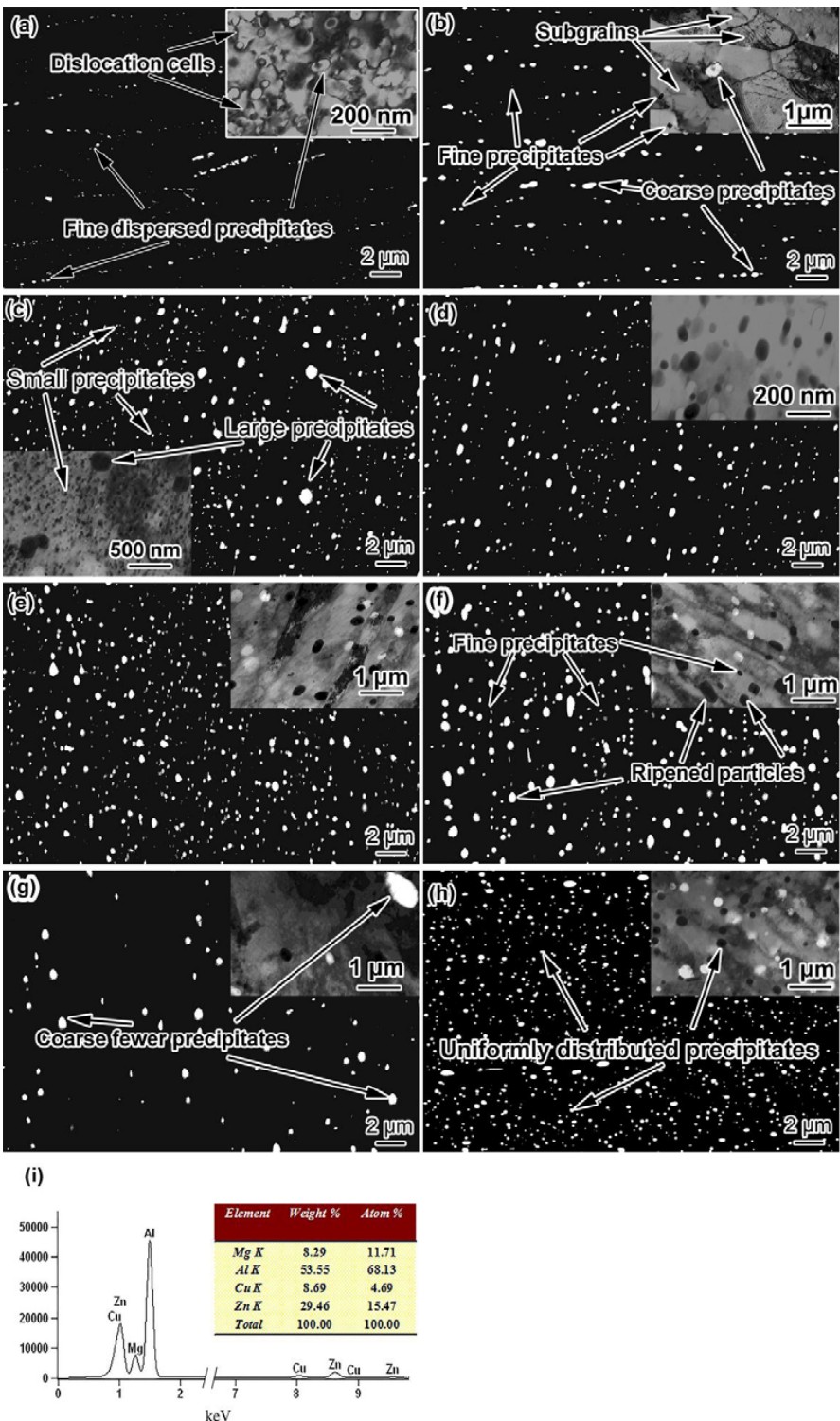

**Figure 5.** SEM microstructures (by backscattered electron) with embedded TEM microstructures: (**a**) DRTMT-60%; (**b**) DRTMT-IA; (**c**) DRTMT$_1$; (**d**) DRTMT$_2$; (**e**) DRTMT$_3$; (**f**) DRTMT$_4$; (**g**) DRTMT$_5$; (**h**) CRTMT-80% and (**i**) EDS result of the white precipitates and XRD spectrum after the 1st-stage deformation.

Commonly, the pinning force $P_Z$ will be enhanced with smaller size and more area fraction of precipitates, which can be expressed as [30]:

$$P_Z = 3F_V\gamma_b d^{-1} \tag{3}$$

where $P_Z$ denotes Zener drag force, N, $F_V$: area fraction of precipitates, %, $\gamma_b$ a constant, N·μm, and $d$ the average particle diameter, μm.

It can be seen that the larger area fraction and the smaller size of precipitates result in higher drag force. Therefore, the migration of grain (sub-grain) boundaries or dislocations would be suppressed or even inhibited so that plenty of dislocation cells were formed during first stage deformation.

The precipitates sizes are not the same during precipitation process. Smaller particles dissolve and larger particles continue to grow, so the average particle size increases and that is called Ostwald coarsening [31]. A similar phenomenon occurs after intermediate annealing process as shown in Figure 5b (some precipitates grow while others are relatively smaller). At the same time, the dislocation cells that were originally entangled became clear sub grains after IA treatment. Precipitates follow the Ostwald coarsening mechanism, and some of them grow up or dissolve back into the matrix, so the pinning effect reduced. Therefore, clearly defined sub grains were formed due to the weakness of drag force and static recovery.

SEM/TEM images of precipitates after the second stage deformation are shown in Figure 5c–g. For DRTMT$_1$, after the second stage deformation the average precipitate sizes were decreased and two different sizes (small: ~50 nm, large: ~300 nm) were formed (Figure 5c). The coarse precipitates were generated by IA while the finer ones were precipitated during the second stage deformation at 300 °C. Also, two size precipitates appear in the DRTMT$_2$, both obey Ostwald ripening process (Figure 5d). As shown in Figure 5e, in DRTMT$_3$, uniformly distributed spherical precipitates and few rod-like particles (~80 nm in transverse and ~200 nm in length) appeared. After the second stage deformation of DRTMT$_4$, the fine spherical DEPs grew to larger sizes (Figure 5f). Contrary to the increasement of particle sizes, the area fraction of the precipitates after second stage deformation decreased markedly compared to that after the first stage deformation. The solubility of (Mg and Zn) would be increased at higher temperature resulting in lower contents of undissolved Mg and Zn (atomic segregated to be MgZn$_2$). So, the equilibrium fraction of precipitates decreased. With increasing temperature, some particles were dissolved into Al matrix to promote others growing/coarsening (Ostwald ripening). So, the density of precipitates is also decreased after the second stage deformation at 430 °C. When the second stage deformation temperature rises to 450 °C (DRTMT$_5$), the solid solubility of MgZn$_2$ in Al matrix increases. And the atomic diffusion rate is increased, so plenty of MgZn$_2$ particles are re-dissolved into Al matrix, resulting in rapid coarsening of the remaining precipitates and sharp reduction of the area fraction (Figure 5g). Compared with DRTMT, the precipitate sizes and distributions of CRTMT are more uniform, which may be related to the consistency of deformation temperature and process. Figure 5i shows the EDS result and these precipitates in SEM images are proved to be MgZn$_2$ phase.

Figure 1 shows the Al-MgZn$_2$ pseudo-binary system, it can be seen that the solubility of MgZn$_2$ phase in Al matrix is increased slowly with raising temperature (<300 °C) but rapidly above 300 °C. Thus, the solubility of MgZn$_2$ in the Al matrix is lower under lower deformation temperature ($T < 300$ °C), which means obvious precipitates appears. Furthermore, for the relatively slow atomic motion and diffusion under lower temperature, the precipitation would be insufficient in a short time (rolling time) from the perspective of dynamics. Moreover, at a relatively low deformation temperature, the higher deformation resistance maybe facilitated the sheet cracking. However, the solubility of MgZn$_2$ will be sharply increased at higher deformation temperature ($T > 300$ °C) greatly weakening the pinning action (refer to formula (3)). Therefore, we set the pre-deformation temperature at 300 °C, which can not only obtain low deformation resistance but also rapid precipitation.

Figure 5i is the XRD spectrum after the first stage deformation. The height of the peak represents the intensity of precipitates (fraction). Kumar et al. [32] and Buha et al. [33] investigated the precipitation

in 7xxx Al alloy and revealed that in typical 7xxx series Al alloys, the interval temperatures, 20–120 °C, 120–250 °C and 150–300 °C, correspond to the formation of GP zones, η' and η phases, respectively, and the interval temperatures 50–150 °C, 200–250 °C, and 300–420 °C, correspond to their dissolution, which is consistent with the statistics of particle area fractions and XRD results (Figure 5i). This confirms that these precipitates in Figure 5 are mainly $MgZn_2$ particles according to the phase calibration. Which is consistent with the result by EDS in Figure 5i. Simultaneously, few T-phase, Fe-rich phase, or S-phase may be formed during the hot rolling, but they cannot be detected by XRD tests for their relatively smaller area fraction.

After first stage deformation, $MgZn_2$ peak appeared which means hexagonal $MgZn_2$ particles were obtained during the first stage deformation at 300 °C (Figure 5a,i). Then, the diffraction intensity of $MgZn_2$ peak decreased a lot indicating area fraction reduction during IA treatment (Figure 5b,i). In other words, $MgZn_2$ particles were largely dissolved into Al matrix (consistent with the TEM/SEM results in Figure 5, decrease of precipitate density and partial coarsening, partial smaller). $MgZn_2$ particles were largely dissolved into Al matrix for the high IA temperature (430 °C) closing to the solid solution temperature (475 °C). Therefore, drag force of precipitates after IA would be weaker than that after first stage deformation consisting with the trends in Figure 5.

Similarly, after 60% hot deformation, the diffraction intensity of $MgZn_2$ peak also appeared in CRTMT alloy, as shown in the Figure 5i. In addition, the peak intensity of CRTMT-60% alloy is between DRTMT-60% and DRTMT-IA, indicating the fraction of precipitates. As shown in Figure 1, the solubility of $MgZn_2$ in Al matrix is greater at 430 °C than that at 400 °C, so more $MgZn_2$ particles is dissolved into Al matrix at 430 °C, resulting in less $MgZn_2$ precipitates. That is why the peak intensity of CRTMT-60% is higher than that of DRTMT-IA. In the same way, the solubility of $MgZn_2$ in Al matrix is lower (Figure 1) and the precipitation tendency is greater when deformed at 300 °C, so the peak intensity of CRTMT-60% is lower than that of DRTMT-60%.

Figure 6a1,b1,c1 shows OM images of grain structures. After the first stage deformation (pre-deformation), all the grains are elongated. Black particles distributed along grain boundaries might be the etched traces of coarse second phases, e.g., $Al_7Cu_2Fe$, $Al_2CuMg$, which cannot be dissolved during the solution treatment. These undissolved, coarse second particles along the grain boundary were crushed during deformation, which might be good for mechanical properties. After 60% deformation, grain spacing becomes narrow (for weaker recovery) in the DRTMT sample (Figure 6b1) with clear grain interior when compared to that of the CRTMT sample (Figure 6a1). The lower pre-deformation temperature of DRTMT-60% (300 °C) as well as good pinning effect may contribute to weaker dynamic recovery and higher deformation storage energy than that of CRTMT-60%. After IA treatment, the static recrystallization apparently occurred and some fine equiaxed grains were formed (Figure 6c1).

Figure 6a2–c2 shows that LAGBs are dominant in the first stage processed alloys, exhibiting a meta-stable micro-structure. The elongated grain boundaries are the HAGBs (in black). The LAGBs (in gray) have low misorientations and cross the elongated grain interiors (Figure 6a2–c2). After 60% warm rolling at 300 °C, grains of DRTMT-60% are narrow (Figure 6b2). At lower rolling temperature, the deformation storage energy obtained after DRTMT-60% would be higher than that after CRTMT-60%. However, these elongated grains are still stretched after DRTMT-IA with some fine equiaxed grains (Figure 6c2). For the CRTMT Al alloys in Figure 6a1,a2, elongated grains with straighter grain boundaries, larger grain spacing, and fewer gray LAGBs appeared, which indicates obvious dynamic recovery occurred during CRTMT-60%.

Figure 6c1–c3 shows the misorientation angle statistic of CRTMT-60%, DRTMT-60%, and DRTMT-IA. The percentage for LAGBs of CRTMT-60%, DRTMT-60%, and DRTMT-IA are 60.11 ± 1.3%, 64.78 ± 1.2%, and 65.95 ± 2.3% respectively. The deformation temperature of DRTMT-60% process is lower than that of CRTMT-60%, more defects (dislocations, etc.) can be accumulated under the same deformation amount, and more substructures (gray lines in Figure 6a2,b2) can be formed by dynamic recovery, so LAGBs of DRTMT-60% are more than those of CRTMT-60% (Figure 6a3,b3).

After the short-time intermediate annealing, DRTMT-IA undergoes a static recovery process, forming a large number of sub-grains (Figures 5b and 6c2), LAGBs slightly increases.

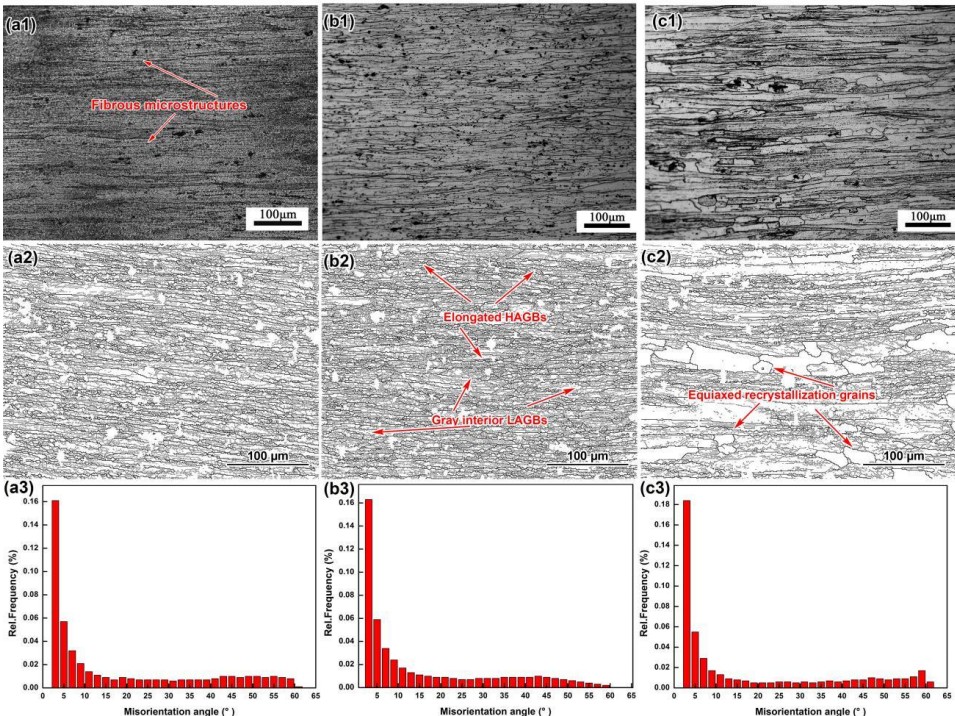

**Figure 6.** OM images (**a1–c1**), EBSD (grain boundaries (**a2–c2**)) maps and grain misorientation angle (**a3–c3**): (**a1–a3**) for CRTMT-60%; (**b1–b3**) for DRTMT-60%; (**c1–c3**) for DRTMT-IA.

### 3.3. The Role of the 2nd-Stage Deformation Temperature in TMTs

After first stage treatment, followed by the second stage deformation. If the second stage deformation temperature is too low (<300 °C), the diffusion rate of atoms will slow down, resulting in insufficient activity for boundaries to transform/migrate. Furthermore, it is easy for the plate to crack when deformed seriously at low temperature. However, if the final hot rolling temperature is too high (>450 °C), according to Al-MgZn$_2$ system in Figure 1, the solubility of MgZn$_2$ increase significantly and almost all the precipitates will be dissolved into Al matrix. It will greatly reduce the pinning effect of DEPs, and weaken grain refinement effect. What's more, temperature of plates will be increased for the deformation heats that maybe causing over burning. Therefore, temperature among 300–450 °C are taken as the selected final hot rolling temperature and the corresponding evolution of precipitation and substructures are also investigated.

Figure 7 shows the TEM and EBSD results of all processed alloys after second stage deformation. Unremarkable dynamic recovery occurred under low second stage deformation temperature of DRTMT$_1$ leading relatively clear grain interior (with fewer gray LAGBs (Figure 7b2)). The distance between elongated grains is short with tangled dislocations (Figure 7b1). When the precipitates grow up to several tens of nanometer in the DRTMT$_2$ sample, they can still pin the movement of dislocations to form dislocation cells or walls while some coarse MgZn$_2$ particles (~300 nm) are precipitated on grain boundaries (Figure 7c1). The energy fluctuation of grain boundary is high, so it is easy to be preferential nucleation site for precipitation. Then, raising second stage deformation to 400 °C (DRTMT$_3$), sub-grains with clear boundaries appeared. Some MSBs (micro sheer bands composed of fine sub-grains arranged according to certain rules, which is transformed from the deformation zone) can be observed after DRTMT$_3$ (Figure 7d1). MSBs can refine sub-grains and perform as preferential nucleation sites in the subsequent recrystallization. At the same time, fine equiaxed grains appeared (Figure 7d2) indicating the transformation of LAGBs to HAGBs (Figure 7d). Sub-structures during the

final hot rolling of DRTMT$_3$ (Figure 5b) will be changed (the activity of boundaries increased gradually, but they are also pinned by DEPs, so the LAGBs are stabilized gradually and transformed into HAGBs) with accumulated larger deformation (80%) and higher temperature (400 °C). Thus, the LAGBs fraction gradually reduced (63 ± 0.9%). The specific grain refinement mechanism will be detailed below. After second stage deformation at 430 °C (DRTMT$_4$), the boundaries were increasingly aligned along RD with the rolling plane and lamellar structures developed (Figure 7e2). Grains are markedly refined to less than 1 µm in transverse after second rolling, and MSBs can also be observed (Figure 7e1). However, when the second stage deformation temperature rises too high (450 °C), most of the DEPs are dissolved into Al matrix and remain seriously coarse (Figures 5g and 7f1), resulting in a sharply decreased area fraction (Figure 8). According to Equation (3), the drag force of precipitates becomes weak. Then, the dislocations/boundaries will be more active. Meantime, a larger lamellar width among 20–65 µm (Figure 7f1,f2) appears for serious dynamic recovery and a lack of pinning effect.

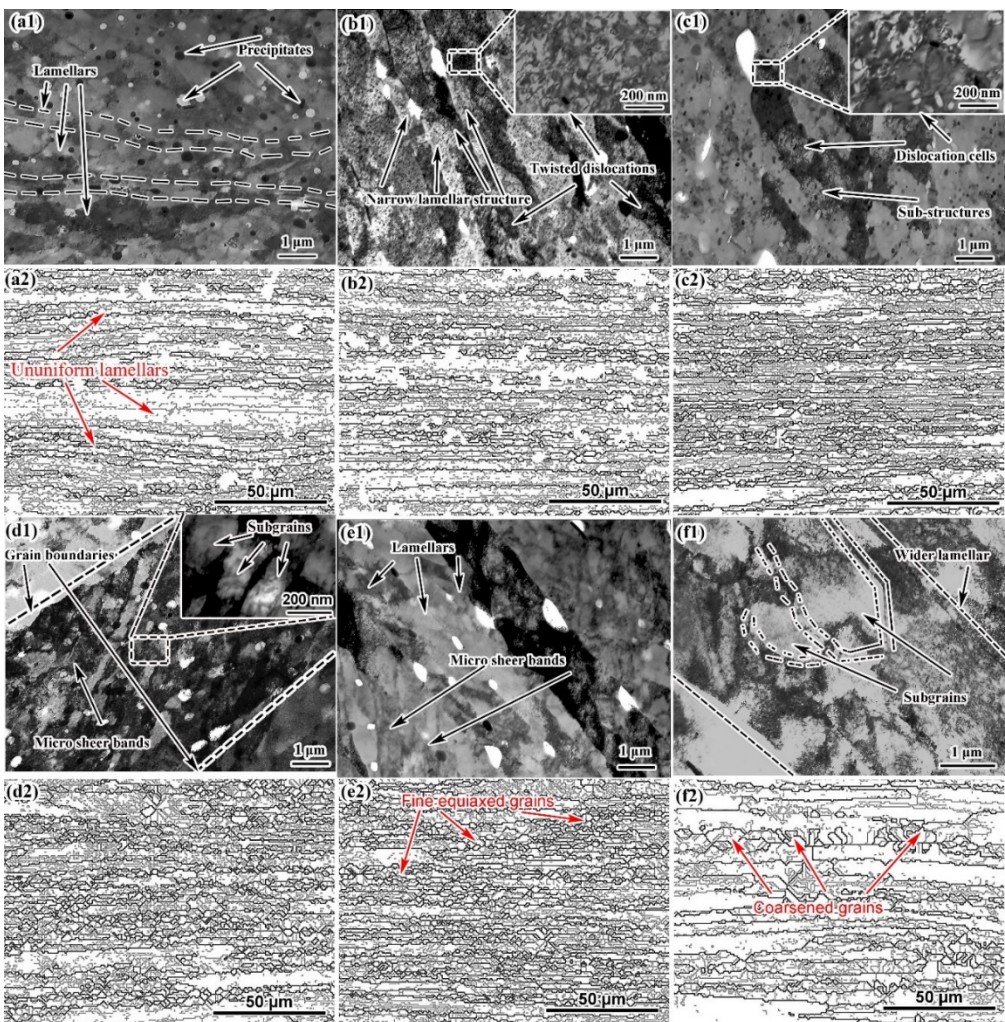

**Figure 7.** TEM ((**a1–f1**) with the enlarged inserted images of the labeled areas) and EBSD ((**a2–f2**) with black HAGBs and gray LAGBs) of 7055 Al alloys after the final hot deformation: (**a1,a2**) for CRTMT; (**b1,b2**) for DRTMT$_1$; (**c1,c2**) for DRTMT$_2$; (**d1,d2**) for DRTMT$_3$; (**e1,e2**) for DRTMT$_4$; (**f1,f2**) for DRTMT$_5$.

Meanwhile, with the increasing second stage deformation temperature, the proportion of LAGBs experienced a fluctuating process. Firstly, in the temperature range of 300–400 °C, the proportion of LAGBs increases (DRTT1: 54.9 ± 1.2%, DRTMT$_2$: 59.1 ± 1.7%, DRTMT$_3$: 64.5 ± 0.9%) with the increasing deformation temperature. In the temperature ranging from 300 °C to 400 °C, with the increasing deformation temperature, the dynamic recovery is intensified, so plenty of sub-grains

(whose boundaries are composed of LAGBs) are formed. Thus, the proportion of LAGBs is increased; Then, in the temperature ranging from 400 °C to 430 °C, with increasing deformation temperature, the proportion of LAGBs decreases (DRTMT$_4$: 59.6 ± 1.1%). In this temperature range, due to the continuous accumulation of deformation at high temperature and the pinning effect of DEPs, some LAGBs gradually stabilized and transformed into HAGBs. Therefore, the proportion of LAGBs decreases with fine equiaxed grains (Figure 7d2,e2). Finally, in the temperature ranging from 400 °C to 430 °C, with the increasing deformation temperature, the proportion of LAGBs increases (DRTMT$_5$: 64.8 ± 1.5%). When the deformation temperature is too high (>430 °C), precipitates are coarsening rapidly and their area fraction decreases sharply directly leading to the weakness of their pinning effect. Finally, with the increasing dynamic recovery and activity of boundaries, plenty of sub-grains were produced, so the proportion of LAGBs increased.

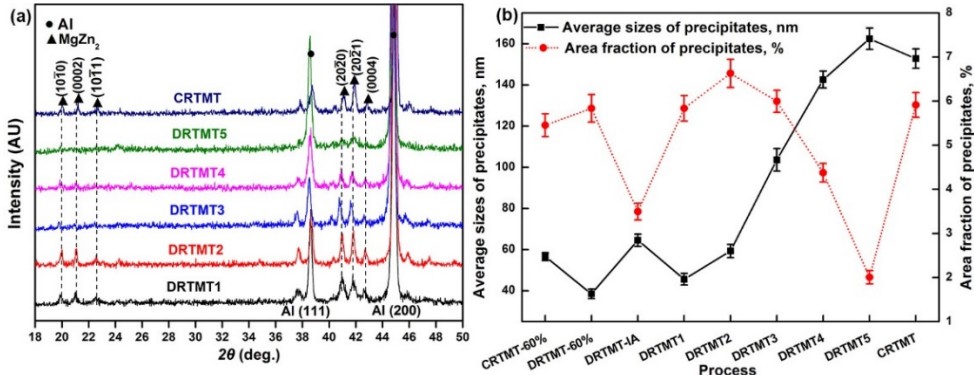

**Figure 8.** (**a**) XRD spectrum; (**b**) the average sizes and fractions of precipitates after the final hot deformation.

The EBSD/TEM results of CRTMT alloy are shown in Figure 7a1,a2. All persistent boundaries, such as HAGBs, will undergo rigid body rotation during deformation at 400 °C, and will eventually be eventually aligned with the rolling plane forming uneven Lamellar microstructures. The widths of lamellar-like grains are quite ununiform ranging from 50 μm to hundreds of nanometers in CRTMT alloy.

Figure 8 shows the XRD spectrum and statistics of precipitates (size and area fraction of precipitates in SEM calculated by software "Image J"). The average size of the precipitates increases with the increasing second stage deformation temperature (Figure 8b). Deschamps et al. found that the coarsening rate of MgZn$_2$ during deformation of 7xxx Al alloy can be expressed as [34]:

$$\frac{\mathrm{d}R}{\mathrm{d}t} = \frac{D}{R} \frac{C_S^\alpha - C_S^{\alpha/P}}{C_S^P - C_S^{\alpha/P}} \frac{C_v}{C_v^{\mathrm{Eq}}} \tag{4}$$

where $C_v$ can be expressed as:

$$C_v = \chi \frac{\sigma\Omega}{Q_f} \varepsilon \tag{5}$$

where $D$ is Diffusion coefficient of atom, $R$ is the radius of globular precipitates, $C_S^P$: the concentration of solute atoms in precipitated particles, $C_S^\alpha$ the concentration of solute atoms in matrix, $C_S^{\alpha/P}$ the concentration of solute atoms in the transition zone between matrix and precipitated particles, $C_v^{\mathrm{Eq}}$ the concentration of equilibrium vacancy, $C_v$ the current vacancy concentration, $\chi$, constant, $\Omega$, the atomic volume, constant, $Q_f$ the vacancy formation energy, constant, $\sigma$ the rheological stress during material deformation, approximate constant, and $\varepsilon$, strain. For 7055 Al alloy, $C_S^\alpha$, $C_S^P$, $C_S^{\alpha/P}$, and $C_v^{\mathrm{Eq}}$ can be approximated as constants. It can be seen that the growth rate of precipitates in the Al alloy is determined by the atomic diffusion coefficient $D$ when the strain $\varepsilon$ is constant. The higher temperature

the greater diffusion coefficient $D$ is and the faster growth rate of precipitates is. So, the average size of the precipitates increases with the increasing second stage deformation temperature. On the other hand, when the deformation temperature is stated, the growth rate of precipitates is determined by the strain $\varepsilon$. The larger $\varepsilon$ is, the faster growth rate of precipitates, resulting in larger precipitates of CRTMT-80% than CRTMT-60% (Figure 8b).

Different from the size evolution, With the increasing second stage deformation temperature, the area fraction of precipitates increases first and then decreases (Figure 8b). This may be related to the solubility and precipitation rate of at different temperatures. The atomic activity is increased by raising temperature, leading to an increase of diffusion/precipitation rate. So, the area fraction of precipitates is increased. However, with the increasing temperature, the solubility of $MgZn_2$ in Al matrix increases, leading to plenty of atoms back dissolving and the amount of precipitates becomes low with the area fraction decreasing to a certain extent. The equilibrium point of the two factors is reached at 350 °C, so the area fraction of precipitates by $DRTMT_2$ is the highest, consistent with the variation tendency (the peak intensity on behalf of the $MgZn_2$ content) of XRD results (Figure 8a) and the SEM/TEM results (Figure 5).

In order to obtain fine-grained microstructures, the suitable second stage deformation temperature may be 430 °C, the lamellar structure could be refined to (0.5–1.2) μm in transverse with micro sheer bands (Figure 7e1) and a large amount of equiaxed fine grains (Figure 7e2). With lower deformation temperature, the slow dynamic recovery and low activity of boundaries make the transition from LAGBs to HAGBs difficult, and the final grain refinement is less significant. (Figure 7b2–d2). However, if the deformation temperature is higher, precipitates are coarsened rapidly and their area fraction decreases sharply (Figure 5g, Figure 7f1,f2 and Figure 8b). According to formula (3), drag force of precipitates becomes weaker. Due to the lack of the drag force, the transformation from LAGBs to HAGBs is difficult, and grains grow faster. Therefore, grain refinement is less obvious under high second stage deformation temperature. In conclusion, 430 °C may be the optimal balanced second stage deformation temperature to obtain effective grain refinement by dislocation rearrangement and LAGBs migration, in turn due to the pinning effects of DEPs.

### 3.4. Fine Grain Mechanism Modeling

Grain refinement is achieved by dislocation rearrangement and LAGBs migration, and by virtue of the pinning effect of DEPs. Figure 9 shows in detail how the grains in DRTMT are refined by analyzing the present results. First, fine DEPs wrapped by high-density dislocations were precipitated during the first stage deformation, which can hinder the slippage of dislocations. Therefore, dislocations were tangled to form dislocation walls/cells (Figure 5a). Then, during the subsequent IA treatment, some DEPs coarsened, while others become smaller or re-dissolved to Al matrix (following Ostwald coarsening rule), and their area fraction decreases (Figures 5b and 8), resulting in weak pinning effect. Therefore, clearly defined sub-grains (formed by LAGBs) appeared out of the dislocation walls/cells (Figure 5b), which increases the proportion of LAGBs. On the other hand, recrystallization during IA treatment also decreases the proportion of LAGBs (Figure 6c1–c3), so it remains nearly the same as DRTMT-60%. Finally, the second stage deformation is carried out and further elongated grain/sub-grain boundaries. DEPs can still impede the motion of boundaries and accelerate the transformation from LAGBs into HAGBs, resulting in decreased LAGBs proportion. Finally, grain refinement can be achieved. (Figure 7d2,e2).

The LAGBs proportion of DRTMT-60% alloy is higher than CRTMT-60%, for defects (vacancies and dislocations) introduced by DRTMT-60% is more than CRTMT-60% with lower deformation temperature. However, the LAGBs of $DRTMT_3$-80% (64.5 ± 0.9%) is a lot less than that of CRTMT-80% (71.79 ± 2.3%) after the final rolling (the same second stage deformation temperature 400 °C and total deformation 80%). For the transition from LAGBs to HAGBs during DRTMT, the proportion of LAGBs is reduced significantly of DRTMT.

In summary, DRTMT provides a new approach of grain refinement for the heat-treatable strengthening alloy including Al-Zn-Mg-Cu alloys. Limited to the capacity of the experimental mill and the inability of the roll to preheat, isothermal rolling cannot be achieved. Moreover, it is difficult to control the rolling parameters precisely. In the future, it is expected to realize isothermal rolling by controlling rolling temperature, thus making full use of DEPs to control rolling structures and realize formation with refined grains of larger and thicker plates.

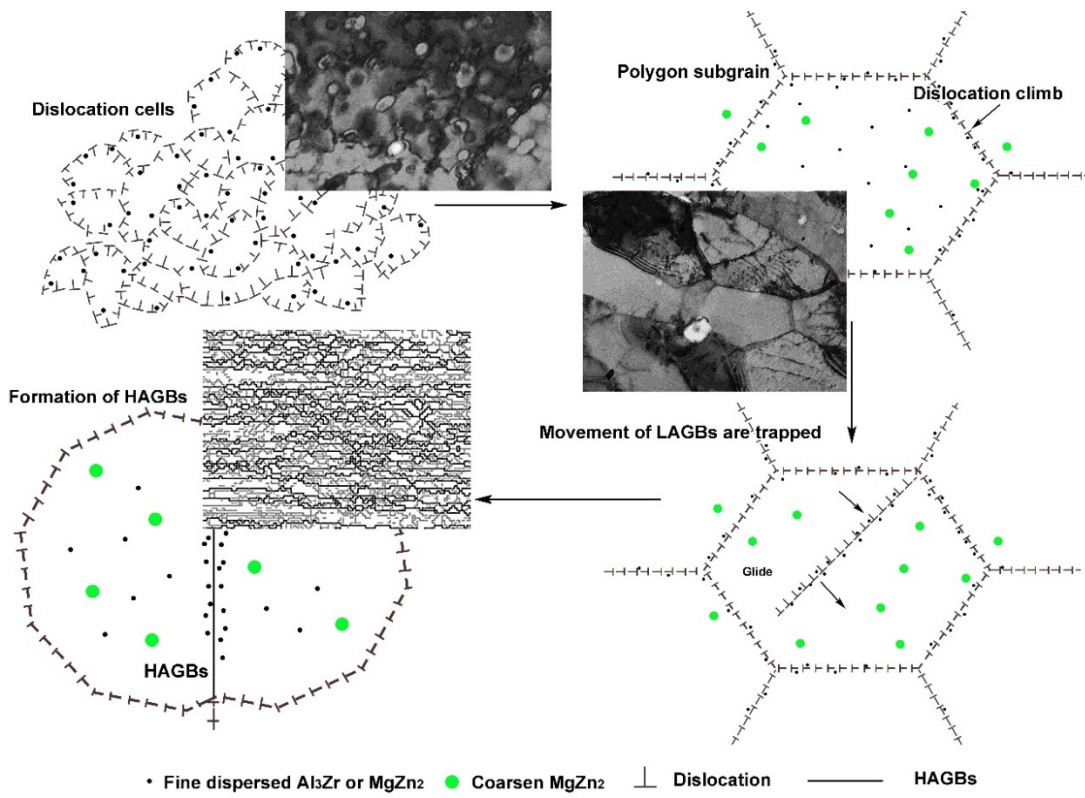

**Figure 9.** Schematic diagram of DRTMT.

## 4. Conclusions

The proposed short cycled DRTMT was applied to manufacture fine grained 7055 Al alloy sheets. The new DRTMT is more energy/time saving than the conventional TMTs with finer grains and better plasticity. Precipitation/boundaries evolution and mechanism for grain refinement during TMTs were investigated.

Plenty of fine DEPs wrapped by dislocation cells were introduced though the first stage deformation (300 °C/60% pre-deformation). Dislocation cells of varying shape and size were formed for dynamic recovery and the drag force caused by DEPs. Partial fine precipitates were re-dissolved into the matrix or merged to be coarsened, inducing the weakening or disappearance of their pinning effects in the subsequent IA process. Polygon sub-grains can be formed out of these original dislocation cells due to the weakness of drag force and static recovery. Finally, the second stage deformation further introduced dislocations. Once the motion of boundaries is impeded by precipitates, the LAGBs into HAGBs transition will be accelerated, and new fine grains can be formed so as to obtain effective grain refinement. The plastic of DRTMT is significantly improved while maintaining high strength when compared with the CRTMT after T6 treatment.

**Author Contributions:** Conceptualization, J.Z. (Jinrong Zuo), L.H. and J.Z. (Jishan Zhang); methodology, J.Z. (Jinrong Zuo) and W.P.; formal analysis, J.Z. (Jinrong Zuo); investigation, J.Z. (Jinrong Zuo), X.S. and A.Y.; writing—original draft preparation, J.Z. (Jinrong Zuo); writing—review and editing, L.H., X.S., W.P., A.Y. and J.Z. (Jishan Zhang); visualization, J.Z. (Jinrong Zuo), X.S., W.P. and A.Y.; supervision, X.S. and J.Z. (Jishan Zhang);

project administration, X.S. and J.Z. (Jishan Zhang). All authors have read and agreed to the published version of the manuscript.

**Funding:** This research is supported by the Natural Science Foundation of Zhejiang (No. LQ19E010003), projects in Science and Technique Plans of Ningbo City (2019B10100), the Major State Research and Development Program of China (No. 2016YFB0300801), the State Key Laboratory for Advanced Metals and Materials of China (No. 2019-Z16), and Sponsored by the K.C. Wong Magna Fund in Ningbo University.

**Conflicts of Interest:** The authors declare no conflict of interest.

## Nomenclature

| | |
|---|---|
| TMT | Thermomechanical Treatment |
| 7055sq | 7055 Al alloy after solution and water quenching |
| CRTMT | Conventional Rolling Thermomechanical Treatment |
| DRTMT | Double-stage Rolling Thermomechanical Treatment |
| DRTMT-60% | 300 °C/60% 1st-stage pre-deformation |
| CRTMT-60% | 400 °C/60% hot rolling |
| IA/DRTMT-IA | Intermediate Annealing after preformation |
| DEP | Deformation-Enhanced Precipitation |
| PSN | Particle-Stimulated Nucleation |
| LAGBs | Low Angle Grain Boundaries |
| HAGBs | High Angle Grain Boundaries |
| T6 | Peak-aging after rolling |
| RD | Rolling Direction |

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
