# Peer review of "Grain Refinement Assisted by Deformation Enhanced Precipitates through Thermomechanical Treatment of AA7055 Al Alloy"

_metals, doi:10.3390/met10050594_

Round 1
Reviewer 1 Report
Grain refinement assisted by deformation induced precipitates through thermomechanical treatment of AA7055 Al Alloy
by : J. Zuo et al.
This paper presents an interesting study of the impact of the grain refinement adjusted by the precipitation of MgZn2 that is tailored by deformation on the elongation (upon deformation) of Al alloys. The acting fundamental mechanisms are cleary described.
While interesting this paper with the DRTMT does not present excessive novelty. In industry nobody would hot-roll with such conditions (CRTMT) right after casting. Besides, Figure 7 shows that some of the proposed DRTMTs induce similar results to a simple CRTMT.
Also, it is not clear from the TEM micrographs that plenty of fine spherical precipitates wrapped by high density dislocations were introduced though the 1st-stage deformation.
Check the writing of authors when citing a paper in the text. (e.g. 'E.A.StarkeJr ' ?)
Abstract:
Authors should be more rigorous with definitions: what do the acronyms 'CRTMT' and 'DRTMT' stand for? an acronym is suposed to correspond to a name, which it doesn't in the first sentence:
'conventional hot rolling (CRTMT) alloy'; I understand 'C' stands for 'conventional', 'R' for 'rolling', but what do the 'T','M' and 'T' correspond to?
'thermomechanical treatment' probably, which comes only in the next sentence.
Equally, in 'DRTMT', what does the 'D' stand for? There is no hint for it in the abstract.
Please clarify.
Same for 'PSN' in the introduction. (acronym definition)
Why CRTMT was not tested also as CRTMT-IA, as this would make a point of comparison to understand the real impact of pre-deformation (as all DRTMT went through IA if I understood correctly) ?
Fig. 3 : h (CRTMT) presents a much better precipitate distribution than g (DRTMT). How come the authors then claim DRTMT is better than CRTMT?
Phase diagram of Fig. 4 should come earlier, could be in the introduction.
Fig. 6: TEM micrographs are of poor quality: too contrasted for some (loss of information in the darks), too weak contrast for others. It is difficult to visualize what is claimed by the authors.
Also, authors claim that TEM micrographs exhibit fine precipitates. I don't see that. Visible precipitates are about 100 nm, which is not fine (e.g. Fig. 6 c1). Besides,
the comparison of Fig. 6 c1 to Fig. 6 a1 (or any other) is not possible because they don't have the same scale.
Please be consistent with the naming of samples, following Table 1. I found:
DRTMT-60%; DRTMT-IA; DRTMT1; DRTMT2; DRTMT3; DRTMT4; DRTMT5; CRTMT-80%;
DRTMT2-7055; DRTMT3-7055; DRTMT-7055; CRTMT; DRTMT4-7055; CRTMT-7055
Please clarify
'DIPs' or 'Deformation induced precipitates': atomic mobility is needed for precipitation to occur. Therefore, to me it is not the deformation that induces the precipitation, but the thermal treatment imposed by the temperature at which the deformation has been performed that induces the precipitation. In other words: if there is not enough atomic mobility there will be no precipitation, even if the dislocation density is very high. 'Deformation induced precipitates' is thus a misleading concept. Please amend.
Author Response
Dear reviewer,
Our manuscript entitled ‘Grain refinement assisted by deformation induced precipitates through thermomechanical treatment of AA7055 Al Alloy’ was submitted (metals-776280) and was revised due to some mistakes or insufficiencies pointed out by reviewers. Thank you very much for giving us the opportunity to revise it. I would like to take this opportunity to thank you for your great work. We greatly appreciate the reviewer’s comments and we have modified the whole manuscript based on the reviewer’s comments carefully. Now we resubmit it and thank you for favorable consideration in advance. We are looking forward to hearing from you. The detailed responses to reviewers are listed as following:
Reviewer’s comments :
1, While interesting this paper with the DRTMT does not present excessive novelty. In industry nobody would hot-roll with such conditions (CRTMT) right after casting.
Answer:
The reviewer must be a very professional expert in this field. Thank you for your valuable suggestions. It is true that there are many production processes and parameters for aluminum alloy plates, so it is difficult to make a comprehensive comparison. For the determination of CRTMT process, we referred to the parameters in [1].
P86: For aluminum alloy plates with thickness less than 6.0 mm, the most basic processes are ingot casting, homogenization, hot rolling, hot finishing rolling, cold rolling and heat treatment. Hot rolling is a continuous process in industry.
P115: Temperature determination: for 7xxx aluminum alloy, the start rolling temperature is 380-410℃, and the finish rolling temperature is 350-400℃.
P116: Hot rolling deformation amount determination: for hot rolling products, the hot rolling deformation shall change the casting structure into rolling structure so as to improve the product properties. The total deformation of aluminum alloy hot rolling products shall be more than 80%.
So we take 400℃/ 80% (CRTMT) as the contrast item to compare with DRTMT.
[1] Zhong L, Ma Y Y, Xie Y C, Production technology of aluminum alloy plate. Metallurgical industry press. Beijing. 2009. P86, 115, 116.
2, Figure 7 shows that some of the proposed DRTMTs induce similar results to a simple CRTMT.
Answer:
Fig. 7. shows the XRD spectrum, the average sizes and fractions of precipitates. Among them, the peak intensity of MgZn2 in CRTMT and DRTMT3 are relatively similar. The height of the peak represents the intensity of precipitates (fraction). This is consistent with the statistical results of the area fraction in Fig.7 (b). According to the Al-MgZn2 phase diagram, the fraction of precipitates is determined by temperature. The final hot rolling temperature of CRTMT and DRTMT3 are both 400℃, so the fraction is similar. However, in addition to the similar fraction of precipitates, the grain structure (Fig. 6 (a2 and d2)) and dislocation configuration (Fig. 6 (a1 and d1)) of CRTMT and DRTMT3 are different, as well as the grain structure after final recrystallization (Fig. 1 (a) and (d)).
3, Also, it is not clear from the TEM micrographs that plenty of fine spherical precipitates wrapped by high density dislocations were introduced though the 1st-stage deformation.:
Answer:The reviewer’ advice is fairly important. In our revised manuscript, we corrected the inappropriate description to “spherical particles wrapped by dislocation cells were precipitated” (line 29 Page 6). Dislocation tangles to form a cellular substructure, and the dislocations in the cell slide towards the cell wall as shown in Fig. 1 (Fig. 5 (a) on Page 8 in our revised manuscript).
Fig. 1 TEM of 300oC/60%
4, Check the writing of authors when citing a paper in the text. (e.g. 'E.A.StarkeJr ' ?)
Answer:
In our revised manuscript, we carefully checked and corrected the author's name.
5, Abstract:
Authors should be more rigorous with definitions: what do the acronyms 'CRTMT' and 'DRTMT' stand for? an acronym is suposed to correspond to a name, which it doesn't in the first sentence: 'conventional hot rolling (CRTMT) alloy'; I understand 'C' stands for 'conventional', 'R' for 'rolling', but what do the 'T','M' and 'T' correspond to? 'thermomechanical treatment' probably, which comes only in the next sentence. Equally, in 'DRTMT', what does the 'D' stand for? There is no hint for it in the abstract. Please clarify. Same for 'PSN' in the introduction. (acronym definition)
Answer:
In the abstract of the revised manuscript, we defined the acronyms in the first sentence. In addition, for the definition of acronyms in the full text, we added a detailed nomenclature table on Page 2.
6, Why CRTMT was not tested also as CRTMT-IA, as this would make a point of comparison to understand the real impact of pre-deformation (as all DRTMT went through IA if I understood correctly) ?
Answer:
The reasons are as follows:
First of all, why do we need IA process for DRTMT? The pre-deformation 300℃/ 60%, deformation temperature is lower, the size of DEPs is smaller, their pinning force is strong resulting in a large number of dislocation cells. The deformation storage energy is higher. Therefore, in the subsequent IA process, a large number of sub grains/ equiaxed grains were formed due to static recovery/ recrystallization. Dislocation rearrangement and grain boundary migration occur in the final hot deformation under such structure to achieve grain refinement. Then let's see the CRTMT process. After 400℃/ 60% , the deformation temperature is higher and the deformation storage energy is lower. Moreover, the temperature of IA process (430℃) is very close to that of CRTMT (400℃). So it is of small significance to treat CRTMT with IA process. Finally, the CRTMT process was designed as a continuous rolling process with only one stage deformation. DRTMT is a separate two-stage rolling process, with short-time annealing between the two stages.
7, Fig. 3 : h (CRTMT) presents a much better precipitate distribution than g (DRTMT). How come the authors then claim DRTMT is better than CRTMT?
Answer:
The reasons are as follows:
First of all, why are the precipitates of CRTMT alloy distributed more evenly and have higher size consistency? The reason is that there is only one rolling stage in CRTMT, the hot deformation temperature is consistent and there is no intermediate treatment. Compared with CRTMT, DRTMT process has extra pre-deformation and intermediate annealing, resulting in two size of precipitates (low size consistency), and due to deformation enhanced precipitation, the distribution of precipitates is not uniform. However, the process and effect of precipitates in the two alloys are different. The effect of drag force of precipitates in CRTMT is lower, and the size is insufficient for PSN (particle stimulated nucleation) effect, so grain refinement is less obvious. For the precipitates (DEPs) in DRTMT, DEPs pinned dislocations to form dislocation cells during pre-deformation, then followed Ostwald ripen during IA to promote the formation of subgrains, finally DEPs pinned boundaries during the final hot rolling to promote the transformation of LAGBs to HAGBs. Therefore, grain refinement is obvious in DRTMT.
Then, does the uniform precipitation of CRTMT affect the final properties of the plate? In the intermediate thermomechanical treatment, the second phase particles play an auxiliary role in the evolution of microstructure and do not directly participate in the aging strengthening of the final material. In the follow-up recrystallization + T6 treatment, these precipitates will re dissolve and re precipitate. The final matirx precipitates in the T6 aging is 5 ~ 7 nm fine dispersed particles. Therefore, the precipitation with uniform distribution of CRTMT here has no effect on the properties of the final material. On the contrary, the fine grains of DRTMT will optimize the grain boundaries precipitates during the final T6 aging. specific contents can be referred to our previous work [1].
- Zuo J R, Hou L G, Shi J T, Cui H, Zhuang L Z, Zhang J S. Effect of deformation induced precipitation on dynamic aging process and improvement of mechanical/corrosion properties AA7055 aluminum alloy. Alloy. Compd2017, 708: 1131-1140.
8, Phase diagram of Fig. 4 should come earlier, could be in the introduction.
Answer:
Thank you very much for your advice. In our revised manuscript, Phase diagram comes earlier in the introduction on Page 1.
9, Fig. 6: TEM micrographs are of poor quality: too contrasted for some (loss of information in the darks), too weak contrast for others. It is difficult to visualize what is claimed by the authors. Also, authors claim that TEM micrographs exhibit fine precipitates. I don't see that. Visible precipitates are about 100 nm, which is not fine (e.g. Fig. 6 c1). Besides, the comparison of Fig. 6 c1 to Fig. 6 a1 (or any other) is not possible because they don't have the same scale.
Answer:
1, In order to make more details visible, we tried our best to readjust the contrast and brightness of TEM image.
Part of the dark areas may be due to tangled dislocation walls or cells. The corresponding part has been explained by embedding enlarged pictures on Page 11. In the TEM experiment, Hitachi H-800 transmission electron microscope of the 90s was used, and pictures were output in the form of film. So the picture quality may be worse than that of modern instruments.
However, the precipitates pinning effect, subgrains, micro sheer bands and lamellar structures we expressed are all visible (marked in the figures on Page 11).
2, The relevant inappropriate description has been modified to replace fine precipitates with precipitates.
3, In Figure 6, we tried to unify the scale. In Figure 5, we also unify the scale of SEM for comparison.
10, Please be consistent with the naming of samples, following Table 1. I found: DRTMT-60%; DRTMT-IA; DRTMT1; DRTMT2; DRTMT3; DRTMT4; DRTMT5; CRTMT-80%; DRTMT2-7055; DRTMT3-7055; DRTMT-7055; CRTMT; DRTMT4-7055; CRTMT-7055 Please clarify
Answer:
We apologize for this mistake. In our revised manuscript, the name of samples are consistent now.
11, 'DIPs' or 'Deformation induced precipitates': atomic mobility is needed for precipitation to occur. Therefore, to me it is not the deformation that induces the precipitation, but the thermal treatment imposed by the temperature at which the deformation has been performed that induces the precipitation. In other words: if there is not enough atomic mobility there will be no precipitation, even if the dislocation density is very high. 'Deformation induced precipitates' is thus a misleading concept. Please amend.
Answer:
The reviewer’ advise is fairly important. If there is not enough atomic mobility there will be no precipitation, even if the dislocation density is very high. I agree with the reviewer. Deformation may accelerate precipitation or affect their appearance, but cannot induce precipitation. So, in our revised manuscript, We use deformation enhanced precipitates (DEPs) instead of deformation induced precipitates (DIPs).
We tried our best to improve the manuscript and made some changes (Marked with red font) in the manuscript. And here we did not list all the changes. We appreciate for editor / reviewer’s warm work earnestly. Once again, thank you very much for your comments and suggestions.
Best regards
Sincerely Yours,
Jinrong Zuo
College of Mechanical Engineering and Mechanics, Ningbo University, Ningbo 315211, P. R. China
E-mail: zuojinrong @nbu.edu.cn (Jinrong Zuo)

Reviewer 2 Report
Please, find the comments in the attachment.

Author Response
Dear reviewer,
Our manuscript entitled ‘Grain refinement assisted by deformation induced precipitates through thermomechanical treatment of AA7055 Al Alloy’ was submitted (metals-776280) and was revised due to some mistakes or insufficiencies pointed out by reviewers. Thank you very much for giving us the opportunity to revise it. I would like to take this opportunity to thank you for your great work. We greatly appreciate the reviewer’s comments and we have modified the whole manuscript based on the reviewer’s comments carefully. Now we resubmit it and thank you for favorable consideration in advance. We are looking forward to hearing from you. The detailed responses to reviewers are listed as following:
- The careful revision of English language should be performed, especially in Introduction and Section3. A lot of sentences begin with the word “which” that is unacceptable. Many sentences are too complex (e.g. “Zuo J R et al [5] designed an improved thermomechanical treatment and found that grain refinement can benefit mechanical properties and Alhamidi A, Horita Z [6] revealed that…”). Some definitions need to be checked (e.g. undissolvable). The misprints need to be corrected (e.g. lamebllar, composesd)
Answer:
Thank you very much for your comments. We revised the manuscript word by word or sentence by sentence to correct the mistakes of grammar and spelling. We had reduced the use of attributive clauses. And the complex sentence was divided into simple sentences to express.
- Thereare a lot of abbreviations that need to be decrypted in the appropriate places (e.g. in Abstract) for ease of understanding.
Answer:
We apologize for our carelessness and revised accordingly. In the abstract of the revised manuscript, we defined the acronyms in the first sentence. In addition, for the definition of acronyms in the full text, we added a detailed nomenclature table on Page 2.
- In Materials and methods: What is meant by initial state in “Which were solution treated ( 470ºC / 16 h + 475ºC / 8 h) to return to initial state”?
Answer:
“initial state” is Al alloy in recrystallized state and most precipitated particles dissolved back. The corresponding explanation has been added to line 11 page 3 with pictures in Fig. 2 on page 4.
- InMaterials and methods: How many passes were applied during the both processes (DRTMT and CRTMT)?
Answer:
There are 7 passes applied during the both processes (DRTMT and CRTMT). Table 2 is added on page 3 to show the specific pass reduction and deformation rate .
- In the Table 1, the one set of parameters of the CRTMT process is mentioned (400ºC/80% 15 -3 mm). However, other parameters (60%) appeared in the further description of the results.Therefore, it is not clear, how many parameters were used in the CRTMT process?
Answer:
CRTMT has only one stage of hot rolling, and the specific process parameters are shown in Table 1 and table 2 (added on page 3). CRTMT-60% was water quenched when plats were rolled to 6 mm (15-12-9-7.5-6). Intermediate sampling for comparison with DRTMT 1st-stage deformation.
- In1 : What is meant by “uniform-distributed microstructures”? Such a term seems to be incorrect.
Answer:
We apologize for this mistake. In our revised manuscript, the corresponding incorrect description has been deleted.
- In Figure 2 (b), is it extremely difficult to see any coarse particles. It needs to be highlighted with EDX analysis or directly in the
Answer:
In our revised manuscript, coarse particles were marked and the corresponding EDX analysis was added in Fig. 4 on Page 6.
- Inthe Section 2: it is claimed that the spherical particles (~ 38 nm) precipitated after the 1st-stage deformation (DRTMT-60%). This question is opened since there isn’t any images of the initial state (before deformation). Therefore, it is impossible to conclude whether these particles form during the 1st deformation stage or during solution treatment that traditionally happens in aluminum alloys. Moreover, all precipitates were determined as MgZn2 phase. It means there aren’t any dispersed phases or intermetallic phases, which were mentioned at the beginning of the chapter?
Answer:
- In our revised manuscript, we added the microstructure characterization of the initial state in Fig. 2 on Page 4. thesphericalMgZn2 precipitates were proved to be formed during the 1st-stage deformation rather than during solution treatment.
- The added Fig. 2 also shows that the hollowed fishbone like particles are iron rich phase while the coarse spherical particles are S phase. These two phases can not be dissolved back in the solution treatment. They are large in size and small in number and remain almost unchanged during heat treatment. In order to better study the evolution of MgZn2precipitates, the later SEM and TEM pictures try to avoid these coarse particles and reduce their field occupation.
Dispersed phases like Al3Zr: the size of Al3Zr is too small as well as their quantity, and the resolution is not enough under this magnification. It is impossible to see the bean shaped Al3Zr in the vision.
- What is meant in the following sentence: “Thus, the solubility of MgZn2 in the Al matrix is lowerunder lower deformation temperature (T < 300ºC) which means obvious precipitates appears.” ?
Answer:
It can be seen from the phase diagram of Al-MgZn2 that with the decrease of temperature, the solubility of MgZn2 in Al matrix becomes lower, forming supersaturated solid solution. At this time, the solid solution is very unstable, and the precipitation of MgZn2 can reduce the supersaturated solid solution, so there is a strong tendency of precipitation , and the volume fraction of MgZn2 precipitates is high in equilibrium state.
- Thereis the sentence “In addition, the peak intensity of CRTMT-60% alloy is the highest after the 1st-stage deformation”. According to Table 1, there is only one stage in CRTMT
Answer:
CRTMT has only one stage of hot rolling. CRTMT-60% was water quenched when plats were rolled to 6 mm (15-12-9-7.5-6). Intermediate sampling for comparison with DRTMT 1st-stage deformation.
- According to the results of XRD analysis, it is claimed that the MgZn2 peak intensity of DRTMT- 60% is also weaker than that of the CRTMT-60%. If all graphs are equally scaled, the MgZn2 peaksof DRTMT-60% seem to be much stronger than the same of the CRTMT-60%. (e.g. (2020) (2021) (0004) )
Answer:
We apologize for our carelessness and revised accordingly. The MgZn2 peak intensity: DRTMT-IA<CRTMT-60%<DRTMT-60%. In our revised manuscript, the corrected analysis was added in line 12~14 on Page 9.
- There is the sentence “in the DRTMT sample (Fig. 5 (a1)”. However, Fig. 5 (a1) is supposed to be for the CRTMT sample. Please, specify
Answer:
We apologize for our carelessness and revised accordingly. In our revised manuscript, this sentence “in the DRTMT sample (Fig. 5 (a1)” was modified to “in the DRTMT sample (Fig. 6 (b1)” in line 19 on Page 9.
- Thereis the sentence “Sub-structures after DRTMT-IA (Fig. 3 (b)) will be changed (the activity of boundaries increased gradually, but they are also pinned by DIPs, so the LAGBs are stabilized gradually and transformed into HAGBs) with accumulated larger deformation (80%) and higher temperature (400ºC)”. It is not clear why the accumulated deformation is 80% and temperature is 400ºC after DRTMT-IA when the first deformation stage before annealing was performed at 60% and 300ºC.
Answer:
We apologize for this mistake. In our revised manuscript, this sentence was modified to “Sub-structures during the final hot rolling of DRTMT3 (Fig. 4 (b)) will be changed (the activity of boundaries increased gradually, but they are also pinned by DEPs, so the LAGBs are stabilized gradually and transformed into HAGBs) with accumulated larger deformation (80%) and higher temperature (400ºC)”.
- It is confusing whether the precipitates coarsen or dissolve when the deformation temperature is too high (>430ºC) since the both statements were mentioned above. If the both take place, where is the border betweenthem?
Answer:
In the period of precipitation, the size of precipitates is not the same. Because smaller particles dissolve and larger particles continue coarsening , the average particle size increases. Such a growth mechanism is called Ostwald maturity, or Ostwald growth (as shown in Fig. 1 below). So, both statements exist in the precipitation process.
Fig. 1 sketch of Ostwald ripening
- InConclusion: “DRTMT is more energy / time saving than the conventional TMTs with finer grains and better mechanical properties”. According to the results of the tensile test, the elongation was increased by the proposed method, not all mechanical
Answer:
In our revised manuscript, the “better mechanical properties” was replaced by “better plasticity” in line 12 Page 15.
- Inconclusion: What is meant in the following sentence: “Finally, the 2nd-stage deformation further elongated grain / subgrain boundaries” ?
Answer:
In our revised manuscript, this This inappropriate sentence was replaced by “Finally, the 2nd-stage deformation further introduced dislocations”
We tried our best to improve the manuscript and made some changes (Marked with red font) in the manuscript. And here we did not list all the changes. We appreciate for editor / reviewer’s warm work earnestly. Once again, thank you very much for your comments and suggestions.
Best regards
Sincerely Yours,
Jinrong Zuo
College of Mechanical Engineering and Mechanics, Ningbo University, Ningbo 315211, P. R. China
E-mail: zuojinrong @nbu.edu.cn (Jinrong Zuo)

Reviewer 3 Report
The paper entitled "Grain refinement assisted by deformation induced precipitates through thermomechanical treatment of AA7055 Al Alloy" presents a high quality of the research. The paper may be published in the present form.
The materials, methods, results, and conclusion are written very well an clear.
The proposed short cycled DRTMT looks promising for application in industry and the paper should received high scientific soundness.
Author Response
Dear reviewer,
Our manuscript entitled ‘Grain refinement assisted by deformation induced precipitates through thermomechanical treatment of AA7055 Al Alloy’ was submitted (metals-776280) and was revised due to some mistakes or insufficiencies pointed out by reviewers. Thank you very much for giving us the opportunity to revise it. I would like to take this opportunity to thank you for your great work. We greatly appreciate the reviewer’s comments and we have modified the whole manuscript based on the reviewer’s comments carefully. Now we resubmit it and thank you for favorable consideration in advance. We are looking forward to hearing from you.
We tried our best to improve the manuscript and made some changes (Marked with red font) in the manuscript. And here we did not list all the changes. We appreciate for editor / reviewer’s warm work earnestly. Once again, thank you very much for your comments and suggestions.
Best regards
Sincerely Yours,
Jinrong Zuo
College of Mechanical Engineering and Mechanics, Ningbo University, Ningbo 315211, P. R. China
E-mail: zuojinrong @nbu.edu.cn (Jinrong Zuo)
Reviewer 4 Report
In the manuscript, the effect of deformation-induced precipitates on grain size and properties of Al 7xxx alloy is studied. The paper merits are high and the work done here is worth publishing, however, the English is unacceptable. There is plenty of grammar errors and it is suggested to ask native speaker to correct the text throughout. Additioinally, Fig. 1 axis captions are difficult to read so they should be enlarged.
Author Response
Dear reviewer:
Our manuscript entitled ‘Grain refinement assisted by deformation enhanced precipitates through thermomechanical treatment of AA7055 Al Alloy’ was submitted (metals-776280) and was revised due to some mistakes or insufficiencies pointed out by reviewers. Thank you very much for giving us the opportunity to revise it. I would like to take this opportunity to thank you for your great work. We greatly appreciate the reviewer’s comments and we have modified the whole manuscript based on the reviewer’s comments carefully. Now we resubmit it and thank you for favorable consideration in advance. We are looking forward to hearing from you. The detailed responses to reviewers are listed as following:
Reviewer’s comments :
- In the manuscript, the effect of deformation-induced precipitates on grain size and properties of Al 7xxx alloy is studied. The paper merits are high and the work done here is worth publishing. However, the English is unacceptable. There is plenty of grammar errors and it is suggested to ask native speaker to correct the text throughout.
Answer:
Thank you very much for your comments. We revised the manuscript word by word or sentence by sentence to correct the mistakes of grammar. We hope this manuscript now can meet the required language level.
- Additioinally, Fig. 1 axis captions are difficult to read so they should be enlarged.
Answer:
In our revised manuscript, we reprocessed Fig. 1. Fig. 1 axis captions are now enlarged to be clear on Page 1.
We tried our best to improve the manuscript and made some changes (Marked with red font) in the manuscript. And here we did not list all the changes. We appreciate for editor / reviewer’s warm work earnestly. Once again, thank you very much for your comments and suggestions.
Best regards
Sincerely Yours,
Jinrong Zuo
College of Mechanical Engineering and Mechanics, Ningbo University, Ningbo 315211, P. R. China
E-mail: zuojinrong @nbu.edu.cn (Jinrong Zuo)
Reviewer 5 Report
The authors proposed short cycled DRTMT to manufacture fine-grained 7055 Al alloy sheets. The new DRTMT is shown to be advantageous than the conventional TMTs with finer grains and better mechanical properties. The authors also investigated the precipitation/boundaries evolution and mechanism for grain refinement during TMTs. The paper is well-written. I recommend it be published after minor revision:
1. Please give the units of each variable in equation (1), and explain what d denotes.
2. Please give the units of each variable in equation (2), is d the same variable as that in equation (1)?
3. The resolution of some figures needs enhancement.
4. Are there any limitations of the proposed method and potential solutions? I suggest the authors add some discussion about possible future enhancement/application of their method.
Author Response
Dear reviewer:
Our manuscript entitled ‘Grain refinement assisted by deformation enhanced precipitates through thermomechanical treatment of AA7055 Al Alloy’ was submitted (metals-776280) and was revised due to some mistakes or insufficiencies pointed out by reviewers. Thank you very much for giving us the opportunity to revise it. I would like to take this opportunity to thank you for your great work. We greatly appreciate the reviewer’s comments and we have modified the whole manuscript based on the reviewer’s comments carefully. Now we resubmit it and thank you for favorable consideration in advance. We are looking forward to hearing from you. The detailed responses to reviewers are listed as following:
Reviewer’s comments :
- Please give the units of each variable in equation (1), and explain what d denotes.
Answer:
In our revised manuscript, the units of each variable in equation (2) were added in line 21~22 on Page 5. d denotes the the average grain diameter which also explained in line 18 on Page 5.
- Please give the units of each variable in equation (2), is d the same variable as that in equation (1)
Answer:
In our revised manuscript, the units of each variable in equation (3) were added in line 10 on page 7. d denotes the the average particle diameter different with that in equation (2) (average grain diameter ) which explained in line 10 on Page 7.
- The resolution of some figures needs enhancement.
Answer:
In our revised manuscript, we reprocessed the picture and enhanced the resolution of Fig. 1, 3, 4, 6.
- Are there any limitations of the proposed method and potential solutions? I suggest the authors add some discussion about possible future enhancement/application of their method.
Answer:
we added some discussion about possible future enhancement/application of our method in 9~13 on Page 15
Application: The DRTMT provides a new approach of grain refinement for the heat-treatable strengthening alloy including Al-Zn-Mg-Cu alloy, while improving the mechanical properties.
Limitations and future enhancement:Limited to the capacity of the experimental mill and the inability of the roll to preheat, isothermal rolling cannot be achieved. Moreover, it is difficult to control the rolling parameters precisely. In the future, it is expected to realize isothermal rolling by controlling rolling temperature. Making full use of DEPs to control rolling structures and realize the rolling of larger and thicker plates
We tried our best to improve the manuscript and made some changes (Marked with red font) in the manuscript. And here we did not list all the changes. We appreciate for editor / reviewer’s warm work earnestly. Once again, thank you very much for your comments and suggestions.
Best regards
Sincerely Yours,
Jinrong Zuo
College of Mechanical Engineering and Mechanics, Ningbo University, Ningbo 315211, P. R. China
E-mail: zuojinrong @nbu.edu.cn (Jinrong Zuo)
